# Multimodal Knowledge Graph Error Detection with Disentanglement VAE and Multi-Grained Triplet Confidence

Submission Id: 174*

## Abstract

Multimodal knowledge graphs inevitably contain numerous errors due to the lack of human supervision in their automated construction and updating processes. These errors can significantly degrade the performance of downstream applications that rely on them. Existing researches on knowledge graph error detection primarily focus on leveraging graph structural and textual information to identify triplet errors in unimodal knowledge graphs. However, unlike unimodal knowledge graphs, multimodal knowledge graphs also suffer from mismatches between images and their corresponding entities, referred to as modality errors. These modality errors not only hinder the performance of downstream applications but also impede our effective utilization of the abundant complementary information provided by the visual modality for detecting triplet errors. To this end, we introduce a novel task of multimodal knowledge graph error detection (MKGED) in this paper, aiming at simultaneously identifying both modality errors and triplet errors. Given the lack of datasets for evaluating this task, we first establish two comprehensive MKGED datasets. Furthermore, we propose a novel framework, KGDMC, to address the MKGED task. Within KGDMC, we devise a disentanglement modality reconstruction (DMR) module for modality error detection. This module disentangles each original modality representation into two disjoint components: modality-specific representations and modality-invariant representations, leveraging the cross-modality reconstruction process to detect mismatched visual modalities. Additionally, for the triplet error detection, we propose a multi-grained triplet confidence (MTC) module, incorporating local triplet confidence, global structure confidence, and global path confidence, to collaboratively detect mismatched triplets. Extensive experiments on our constructed two datasets demonstrate the superiority of our proposed framework.

## CCS Concepts

• **Computing methodologies → Knowledge representation and reasoning**; **Probabilistic reasoning**; • **Information systems → Data cleaning**.

## Keywords

knowledge graph, multimodal information, error detection

**ACM Reference Format:**
Anonymous Author(s). 2018. Multimodal Knowledge Graph Error Detection with Disentanglement VAE and Multi-Grained Triplet Confidence. In *Proceedings of Make sure to enter the correct conference title from your rights confirmation emai (Conference acronym 'XX)*. ACM, New York, NY, USA, 10 pages. https://doi.org/XXXXXXX.XXXXXXX

## 1 Introduction

Knowledge graphs (KGs), which provide relational information between entities in the form of triplets (head entity, relation, tail entity), play a key role in various knowledge-driven downstream AI systems, such as recommendation systems [20, 48], question answering [21, 33], and information retrieval [36, 50]. Most existing KGs are constructed by extracting factual information from semi-structured or unstructured web sources using heuristics algorithms, such as Freebase [4], DBPedia [2], NELL [8] and Wikidata [37].

Given the exponential growth and incessant flux of web information, the automated construction and updating of large KGs often lack human supervision. This inevitably results in the introduction of a substantial quantity of noisy triplets. For instance, the widely used KG NELL exhibits a precision rate of 74% [7], indicating that approximately 0.6 million of its triplets are inaccurate. These erroneous triplets result in substantial performance deterioration within downstream tasks that rely on them. Thus, there is an urgent need to delve into effective KG error detection, and numerous approaches [3, 13, 23, 29, 31, 45, 51, 52] have been proposed for this task. While these methods have achieved significant progress, they are designed solely for graph structural and textual information.

However, with the rapid evolution of social media platforms, images coupled with textual content have become the most prevalent form of web-based information. This promotes the rapid development of multimodal knowledge graphs but also brings new challenges to the task of KG error detection. As shown in Figure 1, multimodal KGs, akin to their traditional counterparts, harbor a substantial quantity of triplet errors. Illustratively, errors in the information extraction process may lead to erroneous triplets such as (*Adam Pally, nationality, United Kingdom*), and ineffective updates to the KG may lead to another incorrect triplet, (*Dennis Schröder, play for, Los Angeles Lakers*). In addition to triplet errors, multimodal KGs frequently encounter modality errors during the process of their automatic construction. The right part of Figure 1 is an example. Accompanying images often align with the entire text content, rather than being explicitly tied to a specific entity. This results in a modality error where the image of *Lebron James* is incorrectly associated with *Dennis Schröder*.

In this paper, we are the first to focus on these problems and introduce a new task of multimodal knowledge graph error detection (MKGED), accompanied by the construction of two comprehensive datasets for its evaluation. MKGED mainly handles unique challenges as follows: (1) **Lacking labeled data.** Automatically

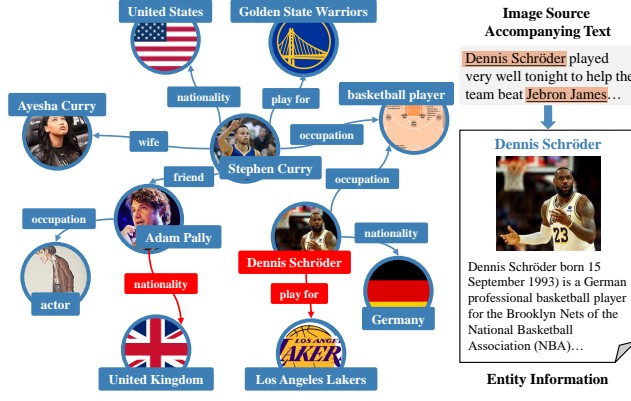

**Figure 1: Examples of some multimodal KG errors. The left part is a subgraph of multimodal KGs, errors in it are denoted in red. The right part is an example of extracting images from web sources to construct multimodal KGs.**

constructed KGs inherently tend to be large and encompass a wide variety of error types. Consequently, the process of manually labeling data is time-consuming and labor-intensive, and difficult to cover all noise patterns. Thus, in real-world practice, MKGED often necessitates the proactive detection of errors within multimodal KGs, without relying on any labeled data during the training phase. (2) **Modality errors.** Given the advanced maturity of information extraction technology, prevalent modality errors encountered in practical multimodal KGs arise from mismatched images that are highly similar or relevant to their associated entities, posing challenges in detecting such modality errors. (3) **Triplet errors.** The distinctive image information within multimodal KGs provides abundant additional complementary knowledge for triplet error detection. However, erroneous image information not only fails to contribute to this process but also introduces additional noise (e.g. *Dennis Schröder* of (*Dennis Schröder, play for, Los Angeles Lakers*)), ultimately diminishing the overall performance.

To address these challenges, we propose a novel multimodal Knowledge Graph error detection framework with Disentanglement VAE and Multi-grained triplet Confidence (KGDMC for short), which utilizes an unsupervised learning way to solve the challenge of lacking labeled data. To detect modality errors, we devise a disentanglement modality reconstruction (DMR) module. This module utilizes cross-modality reconstruction to comprehend noise patterns and examine semantic coherence between modalities. It provides modality confidence for modality error detection. Considering the challenge of high similarity or relevance between images and their erroneously associated entities in modality errors, DMR adopts a disentanglement approach, which disentangles each modality input data into separate modality-specific and modality-invariant feature spaces to obtain more fine-grained semantic information. It then explicitly utilizes accurate textual modality-invariant features to guide the the cross-modality reconstruction process of suspicious visual modality-specific information, ensuring adequate correlation between modalities. To address triplet errors, we propose a multi-grained triplet confidence (MTC) module. MTC tackles the challenge posed by image noise in triplet error detection through the utilization of weighted embedding based on modality confidence. Additionally, MTC employs local triplet confidence, global

structure confidence, and global path confidence to assess the internal factual self-consistency of triplets, the consistency of aggregated neighbor information, and the consistency of information propagation, respectively. The integration of these three distinct granularity levels of confidence significantly enhances our capability to comprehensively detect triplet errors.

The main contributions of this paper are summarized as follows:

- To the best of our knowledge, we are the first to introduce the novel multimodal knowledge graph error detection task, which aims to simultaneously detect both modality errors and triplet errors in multimodal knowledge graphs.
- For modality error detection, we devise a DMR module that leverages disentanglement VAE for cross-modality reconstruction. This approach effectively captures fine-grained semantic consistency across modalities.
- For triplet error detection, we propose a MTC module, which integrates the internal factual self-consistency of triplets, the consistency of aggregated neighbor information, and the consistency of information propagation to collaboratively estimate comprehensive triplet confidence.
- We compare our KGDMC with state-of-the-art baselines on two real-world multimodal knowledge graph datasets, which we construct to incorporate multiple types of errors. Extensive experimental results demonstrate the effectiveness of our proposed framework.

## 2 Related Work

### 2.1 Knowledge Graph Error Detection

Traditional knowledge graph error detection mainly focuses on graph structure and textual information, with this area of research witnessing substantial progress and thorough exploration in recent years. Knowledge graph error detection methods can be roughly divided into two groups: rule-based methods and embedding-based methods. Rule-based methods analyze the commonalities present within triplets through the mining of association rules [1]. Triplets that fail to adhere to these rules are subsequently identified and considered as errors. AMIE [17] formulates a suite of golden rules via path exploration and rule pruning processes. PaTyBRED [31] utilizes heuristic approaches to explore the paths search space and select relevant type and path information for each relation. GRR [10] explores the graph-repairing rules of implication, consistency, and termination. KGIST [3] inductively summarizes knowledge graph to derive a set of soft rules, which describe normal conditions within a KG. However, given the inherent diversity in rules across different KGs and the fact that these methods primarily target specific error categories, their applicability lacks generalizability.

Consequently, most of recent works focus on embedding-based methods, which predict the plausibility between entity and relation representations in low-dimensional vector space. CKRL [45] is the first to introduce triplet confidence, which incorporates both local and global path information, to guide models to pay more attention to convincing triplets. KGTtm [23] integrates entity correlation, relation invariance, and path reachability to predict triplet plausibility. KGClean [18] utilizes active learning to train a classification model to identify and repair erroneous triplets based on a small number of labeled data. CAGED [51] employs contrastive learning to capture

consistency between different triplet hyper-views of a knowledge graph. HEAR [52] leverages the hierarchical path structure within knowledge graphs to detect noise. KAEL [13] assembles a set of advanced KG error detectors through ensemble learning and trains them by active learning to label a small number of samples. CCA [29] utilizes prompt learning to reconstruct textual and structural information to better detect semantically similar noise.

Although these methods have achieved significant progress, they are designed solely for graph structural and textual information, thereby rendering them ineffective in leveraging abundant complementary information of visual modality within multimodal KGs. Additionally, these methods face limitations in addressing modality errors, which frequently occur in multimodal KGs. Therefore, we propose the MKGED task, which aims not only to detect modality errors but also to efficiently utilize complementary visual information to detect triplet errors in the presence of modality noise.

## 2.2 Multimodal Knowledge Graph Embedding

Multimodal knowledge graph embedding (MKGE) typically harnesses triplet structure and multimodal information to embed entities and relations. Subsequently, it defines score functions to measure the plausibility of triplets. IKRL [46] is the first MKGE model, which learns structural and visual information separately based on TransE [5]. Based on IKRL, Mousselly et al. [35] and TransAE [44] integrate visual and structural representations into unified embeddings. RSME [43] utilizes a forget gate to ignore irrelevant image information to obtain better embeddings. Subsequently, MMKRL [30] and OTKGE [6] exhibit enhanced capabilities in aligning multimodal embeddings and integrating multimodal knowledge, facilitated respectively by component alignment and optimal transport. CMGNN [16] employs contrastive learning to achieve multi-modal and high-order structural modeling within graph neural networks. VISTA [27] integrates visual and textual representations of entities and relations by leveraging transformers for entity encoding, relation encoding, and triplet decoding. These MKGE methods have achieved remarkable success in a diverse range of multimodal knowledge graph related tasks, including but not limited to multimodal knowledge graph completion and triplet classification. However, they assume that all triplet facts within multimodal KGs are completely correct, thereby constraining the efficacy of multimodal KG error detection.

## 3 Problem Statement

A multimodal knowledge graph can be depicted or formulated as $\mathcal{G} = (\mathcal{E}, \mathcal{R}, \mathcal{T}, \mathcal{I}, \mathcal{D})$, where $\mathcal{E} = \{e_1, e_2, ..., e_m\}$, $\mathcal{R} = \{r_1, r_2, ..., r_n\}$ respectively denote the sets of entities and relations. $\mathcal{I}$ and $\mathcal{D}$ respectively contains visual images and textual descriptions for all entities. $\mathcal{T} \subseteq \mathcal{E} \times \mathcal{R} \times \mathcal{E}$ is the set of triplets. Each triplet comprises a head entity $h$, a relation $r$, and a tail entity $t$, articulated in the form of $(h, r, t)$. Multimodal knowledge graph errors involve modality errors and triplet errors. For modality errors, given an entity $e_i$ accompanied by its description in $\mathcal{D}$ and image in $\mathcal{I}$, if the image fails to associate with the entity, then the entity $e_i$ is a modality error. Regarding triplet errors within multimodal KGs, for a given triplet $(h, r, t)$, we define the error as the mismatch that arises between its head entity $h$, tail entity $t$, and corresponding relation $r$. Two

distinct types of mismatch can occur: entity mismatch, where the head entity $h$ and tail entity $t$ are irrelevant, and relation mismatch, where relevant entities are erroneously connected through inappropriate relations. For multimodal knowledge graph error detection, given a noisy multimodal KG $\mathcal{G}$, our goal is to rank all entities and triplets according to their respective suspicious scores. These scores, spanning from 0 to 1, serve as indicators of the likelihood of an error existing within the respective entity or triplet. Entities and triplets that exhibit higher scores are designated as errors.

## 4 Dataset Construction

In recent years, multiple real-world multimodal knowledge graph datasets have been widely used. However, these datasets lack explicit labels for errors or noises. Furthermore, with the maturity of multimodal information extraction technology, the errors encountered in real-world multimodal KGs frequently exhibit a high degree of semantic similarity with their accurate counterparts. Consequently, to effectively evaluate the performance of MKGED, we construct datasets by adding various types of noise to simulate the real-world multimodal KGs constructed automatically without human supervision. There are three types for modality errors:

- **Random noise**, originating from a uniform distribution that is independent of the data and inherently unpredictable, is generated by randomly substituting the images.
- **Intra-modality similar noise** simulates real-world modality errors arising from high similarities or correlations among images. To generate them, we first utilize ViT [14] to obtain visual features of all entity images. Subsequently, for a given entity, we compute the distance between its corresponding visual feature and those of other entities. Based on the distance, we select the image exhibiting the highest degree of similarity for replacement.
- **Inter-modality similar noise** simulates modality errors stemming from high semantic similarities between textual content and erroneous images. Given the remarkable modality alignment capabilities of vision-and-language pretrained models, we employ CLIP [32] to extract visual and textual features of images and entity descriptions. Subsequently, for a given entity, we compute the distance between its textual feature and the visual features of other entities to replace with highly relevant images.

For triplet errors, following [29], we also contain three types of noise to simulate real-world multimodal KGs:

- **Random noise** akin to the random noise of modality errors. Given a correct triplet $(h, r, t)$, we randomly substitute either one of the entities or the relation within it to generate noise.
- **Semantic similar noise** mimics real-world triplet errors stemming from high semantic similarities between entities. Given a triplet $(h, r, t)$, we employ BERT [12] to extract textual features of the tail entity $t$ and other tail entities associated with the relation $r$. Then, we calculate the distance between them to replace $t$ by tail entities that exhibit a high degree of semantic similarity.
- **Adversarial noise** is adversarially generated by leveraging knowledge graph embedding models, which are frequently

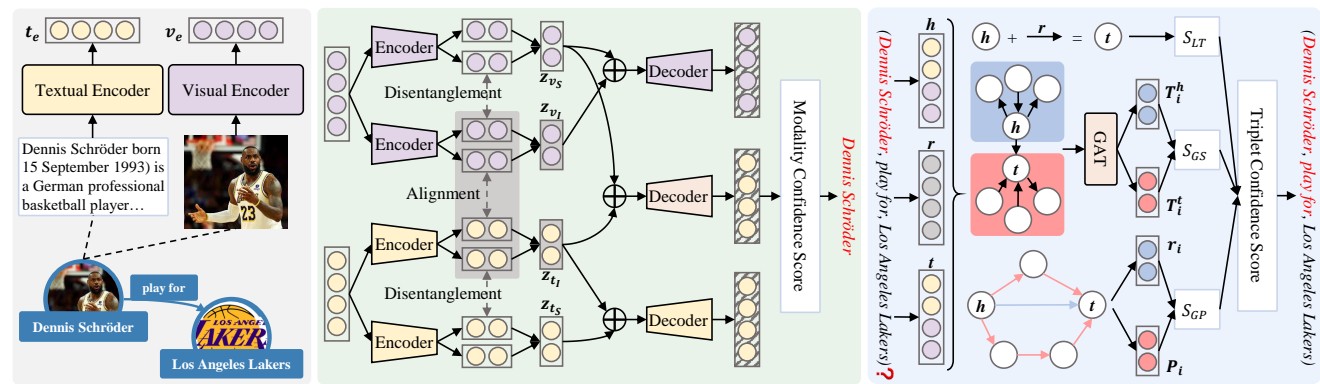

**Figure 2: Overview of our KGDMC framework, which contains three modules: feature extraction, disentanglement modality reconstruction (DMR) for modality error detection, and multi-grained triplet confidence (MTC) for triplet error detection.**

employed to assess the plausibility of triplets. We first generate a substantial quantity of error triplets through random substitution. Then, we train a TransE [5] model to select the most plausible triplets from this set as noise.

## 5 Methodology

Figure 2 illustrates the overall architecture of our proposed KGDMC framework. Initially, we extract textual and visual features for entities in triplets. Subsequently, we utilize disentanglement variational autoencoder (VAE) to decompose each modality into distinct modality-specific and modality-invariant subspaces through distribution alignment and disentanglement constraints. To capture fine-grained semantic consistency between modalities, we explicitly integrate accurate textual modality-invariant information and potentially suspicious visual modality-specific information, guiding the model to cross-reconstruct accurate textual features. This consistency provides the modality confidence scores for detecting modality errors. Based on the modality confidence, we derive entity embeddings by weighting textual and visual features. Finally, we calculate triplet confidence scores from different views with multiple granulations to assess three critical aspects: the internal factual self-consistency of triplets, the consistency of aggregated neighbor information, and the consistency of information propagation. By integrating these scores, we effectively detect triplet errors.

### 5.1 Feature Extraction

To extract meaningful textual features, following [29], we employ pre-trained BERT [12] as our textual encoder. Specifically, for each entity $e_i$, the input sequence is constructed in the following format: [CLS] *title* [SEP] *des* [SEP], where *title* and *des* refer to the word-piece tokens of the entity title and description, respectively. Subsequently, these inputs are fed into BERT to extract textual features $t_{e_i}$, which are the vectors in the last hidden layer corresponding to the position of [CLS] tokens.

To capture expressive visual features, we employ pre-trained Vision Transformer (ViT) [14] as our visual encoder. Given an image associated with the entity $e_i$, we initially resize it to $C \times H \times W$ pixels, where $C$ is the number of channels, and $H \times W$ refers to the image resolution. Then we reshape it into $H \times W/P^2$ flattened two-dimensional patches, where $P$ represents the patch size. Similar

to the textual feature extraction, we feed them into ViT to obtain the hidden state of the [CLS] as our visual features $v_{e_i}$.

### 5.2 Disentanglement Modality Reconstruction

In multimodal KGs, heterogeneous modalities of entities typically encompass diverse aspects of information (e.g. visual line and color, textual detailed description) alongside shared aspects of information (e.g. the characteristics and categories of entities). Disentangling modalities can empower the model to delve into fine-grained semantic nuances, thereby facilitating the acquisition of semantic consistency across modalities. Therefore, we utilize disentanglement VAE [22], which is the extension of VAE [26], for disentanglement.

*5.2.1 Disentanglement VAE.* A standard VAE is typically dissected into two components: an encoder, which extracts a low-dimensional latent variable $z$ from the input data $x$, and a decoder, which subsequently reconstructs an output $\widetilde{x}$ that approximates the original data $x$ based on the latent variable $z$. In VAE, variational inference is employed to find the true conditional probability distribution $p_\phi(z|x)$ over the latent variable $z$. Given the intractability of the true conditional distribution, an approximation way is to find its closest proxy posterior, $q_\theta(z|x)$, by minimizing their distance using a variational lower bound. Thus, the objective function of VAE is the variational lower bound on the marginal likelihood of $x$:

$$\mathcal{L}_{VAE}(x) = \mathbb{E}_{q_\phi(z|x)}[\log p_\theta(x|z)] - D_{KL}(q_\phi(z|x)||p(z)) \quad (1)$$

where the initial term represents a reconstruction loss, and the last term is the Kullback-Leibler (KL) divergence between conditional probability distribution $q_\phi(z|x)$ and the prior distribution $p(z)$ of $z$. The prior distribution is predefined as a multivariate Gaussian distribution. The encoder predicts mean $\mu$ and variance $\Sigma$ of the distribution $q_\phi(z|x) = \mathcal{N}(\mu, \Sigma)$ and then the latent variation $z$ is generated by reparameterization strategy [26].

Disentanglement VAE extends VAE by a pair of encoders responsible for modeling modality-specific and modality-invariant information as follows:

$$\mathcal{L}_{DVAE}(x) = \mathbb{E}_{q_{\phi_S, \phi_I}(z_S, z_I|x)}[\log p_\theta(x|z_S, z_I)]$$
$$- D_{KL}(q_{\phi_S}(z_S|x)||p(z_S)) - D_{KL}(q_{\phi_I}(z_I|x)||p(z_I)) \quad (2)$$

where $\phi_S$ and $\phi_I$ represent the respective parameters of the encoder pairs: $E_S$ and $E_I$. The latent variations $z_S$ and $z_I$ denote the

modality-specific and modality-invariant representations, respectively, extracted from the input data $x$. Furthermore, $q_{\phi_S, \phi_I}(z_S, z_I|x)$ characterizes the joint distribution of $z_S$ and $z_I$ conditioned on $x$. We obtain textual and visual disentanglement VAE by replacing $x$ with $t_e$ and $v_e$ respectively. Finally, the objective function of our multimodal disentanglement VAE is formulated as follows:

$$\mathcal{L}_{MDVAE} = \mathcal{L}_{DVAE}(t_e) + \mathcal{L}_{DVAE}(v_e) \tag{3}$$

*5.2.2    Distribution Alignment and Disentanglement Constraints.* A pivotal character of our disentanglement VAE lines in the construction of modality-invariant and modality-specific subspaces. The modality-invariant subspace focuses on information describing the shared aspects across modalities. To achieve this, we add a distribution alignment loss constraint. Following [34], we utilize the Wasserstein distance between the latent multivariate Gaussian distributions. Given two distributions $a$ and $b$, the 2-Wasserstein distance [19] is formulated as:

$$W_{ab} = (||\mu_a - \mu_b||_2^2 + ||\Sigma_a^{\frac{1}{2}} - \Sigma_b^{\frac{1}{2}}||_{\text{Frobenius}}^2)^{\frac{1}{2}} \tag{4}$$

We construct the modality-invariant subspace by minimizing the distance between textual and visual distributions within this subspace: $\mathcal{L}_{DA} = W_{t_I v_I}$. Furthermore, to construct modality-specific subspaces, we design two constraints for the intra-modality disentanglement and inter-modality disentanglement, respectively. These constraints aim to avoid modality-specific representations carrying the information of shared aspects. For intra-modality, our objective is to ensure that disentangled modality-invariant and modality-specific representations are distinct in the latent space. We achieve this by maximizing the distance between disentanglement distributions of each modality: $\mathcal{L}_{Intra_{DD}} = -(W_{t_I t_S} + W_{v_I v_S})$. For inter-modality, we strive to make the modality-specific representations of different modalities as disparate as possible. To this end, we maximize the distance between textual and visual specific distributions: $\mathcal{L}_{Inter_{DD}} = -W_{t_S v_S}$. Finally, the distribution constraint loss of our disentanglement VAE is constructed as follows:

$$\mathcal{L}_{DC} = \mathcal{L}_{DA} + \alpha \mathcal{L}_{Intra_{DD}} + \beta \mathcal{L}_{Inter_{DD}} \tag{5}$$

where $\alpha$ and $\beta$ are hyperparameters to control the strength of intra-modality disentanglement and inter-modality disentanglement.

*5.2.3    Cross-Modality Reconstruction.* Intuitively, features of the same entity without any modality errors are semantically consistent even though they come from different modalities. Therefore, after obtaining fine-grained semantic information of each modality by disentanglement VAE, we cross-reconstruct modalities to capture this consistency for the detection of modality errors. Given the accuracy of textual modality, we utilize textual modality-invariant representation as guidance to reconstruct potentially suspicious visual modality-specific representation into accurate textual features. In contrast to solely relying on visual representations, the incorporation of textual modality-invariant representations significantly enhances the efficiency of cross-modality reconstruction. To this end, we concatenate $z_{t_I}$ and $z_{v_S}$ to feed into cross-modality decoder $D^{cross}$ to reconstruct $t_e$. The modality confidence score is:

$$S_M = ||t_e - D^{cross}(z_{t_I}, z_{v_S})||_2^2 \tag{6}$$

To mitigate the negative effects of erroneous entities and pay more attention to those more convincing entities, we utilize the modality confidence score to generate adaptive pseudo-labels to guide the model training: $C_M(e_i) = 1 - (S_M(e_i) - \min(S_M)/(\max(S_M) - \min(S_M))$. Our cross-modality reconstruction loss is formulated as:

$$\mathcal{L}_{CR} = \sum_{e_i \in \mathcal{E}} S_M(e_i) \cdot C_M(e_i) \tag{7}$$

Finally, the objective function of our DMR module is to jointly optimize the above loss functions as follows:

$$\mathcal{L}_{DMR} = \mathcal{L}_{MDVAE} + \mathcal{L}_{DC} + \mathcal{L}_{CR} \tag{8}$$

## 5.3    Multi-Grained Triplet Confidence

Considering the negative effect of modality errors, we weight textual and visual features to obtain each entity embedding as:

$$w_i = (\max(S_M) - S_M(e_i))/\max(S_M), \; e_i = t_{e_i} + w_i v_{e_i} \tag{9}$$

Given a triplet $T_i$, we obtain its embedding $(h_i, r_i, t_i)$ via Eq. 9 and the textual content of $r_i$. To comprehensively detect triplet errors, unlike previous works that only involve partial views, we integrate multi-grained triplet confidence from three different views.

Local triplet confidence is employed to measure the internal factual self-consistency within triplets. To achieve this, we utilize the TransE [5], where relations are interpreted as translations between head entities and tail entities. The greater alignment a triplet exhibits with the translation assumption, the higher the likelihood of its correctness. Thus, our local triplet confidence score is defined as: $S_{LT}(T_i) = ||h_i + r_i - t_i||_2$.

Global structure confidence is employed to measure the consistency of aggregated neighbor information. Following [51], we regard each triplet as a node and obtain its embedding by concatenating internal embeddings of the triplet: $T_i = [h_i, r_i, t_i]$. Subsequently, we generate two subgraphs by connecting triplets adjacent to head entities $\mathcal{T}^h$ and triplets adjacent to tail entities $\mathcal{T}^t$, respectively. The triplet set $\mathcal{T}^h$ represents triplets that share the same head entity, i.e. $\mathcal{T}^{h_i} = \{T_j|h_j = h_i \vee t_j = h_i\}$. Similarly, the set $\mathcal{T}^t$ is defined as $\mathcal{T}^{t_i} = \{T_j|h_j = t_i \vee t_j = t_i\}$. According to social community theory [40], correct triplets are those individuals who can successfully integrate into society. Therefore, if a triplet is correct, its embeddings aggregated from different neighbors of the two subgraphs should exhibit a high degree of consistency. To measure this consistency, we utilize the graph attention network (GAT) [41] to respectively weighted aggregate neighbors from the two subgraphs to obtain $T_i^h$ and $T_i^t$. Then, our global structure confidence score can be defined as: $S_{GS}(T_i) = ||T_i^h - T_i^t||_2$.

Global path confidence is employed to measure the consistency of information propagation. Between any two entities, multiple paths may exist, each embodying a distinct process of information propagation between them. If a triplet is correct, its relation will be consistent with the comprehensive information conveyed by the other paths. Initially, we utilize the depth-first search (DFS) algorithm to retrieve the paths between head entities and their corresponding tail entities in all triplets. Given that triplets adhere to the translation assumption based on local triplet confidence, we sum up the corresponding relation embeddings to obtain the embedding of each individual path: $p_o = \sum_{j=1}^{l} r_j$, where $l$ is the number of relations within this path. Due to the existence of erroneous triplets, not all paths are reliable. We compute the average

**Table 1: Overall statistics of datasets. E. means the errors.**

| Datasets | Entities | Relations | Triplets | Entity E. | Triplet E. |
|---|---|---|---|---|---|
| FB15K-237 | 14,541 | 237 | 326,437 | 727 | 16,321 |
| WN18RR | 40,943 | 11 | 97,897 | 2,047 | 4,894 |

local triplet confidence along the path to determine the path importance: $AS(p_o) = \sum_{j=1}^{l} S_{LT}(T_j)$. The less likely it is that the path contains incorrect triplets, the more important it is. Then, we obtain the unified path embedding based on this importance: $\boldsymbol{P_i} = \sum_{o=1}^{O} \frac{AS(p_o)}{\sum_{j=1}^{O} AS(p_j)} \boldsymbol{p_o}$, where $O$ is the number of paths between $h_i$ and $t_i$. Similar to global structure confidence, our global path confidence score is defined as: $S_{GP}(T_i) = ||\boldsymbol{r_i} - \boldsymbol{P_i}||_2$.

Finally, we integrate these three confidence scores to determine the ultimate suspicious score of triplets:

$$S_T = S_{LT} + \lambda S_{GS} + \delta S_{GP} \tag{10}$$

where $\lambda$ and $\delta$ are hyperparameters to balance the scores. Similar to modality error detection, we generate adaptive pseudo-labels $C_T$ to guide the training process of triplet error detection. The ultimate loss of our MTC module is calculated as follows:

$$\mathcal{L}_{MTC} = \sum_{T_i \in \mathcal{T}} \sum_{T_i^- \in \mathcal{T}^-} \max(0, \gamma + S_T(T_i) - S_T(T_i^-)) \cdot C_T(T_i) \tag{11}$$

where $\gamma$ is a margin hyperparameter, $\mathcal{T}$ represents all triplets. And $\mathcal{T}^-$ denotes negative triplets generated by randomly replacing head or tail entities, as explicit negative triplets are not available.

## 6 Experiments

In this section, we conduct comprehensive experiments to sufficiently verify the effectiveness of our proposed framework KGDMC. We intend to investigate the following research questions (RQs):

- **RQ1.** How does the KGDMC perform compared with various state-of-the-art baseline methods?
- **RQ2.** How does each individual component of KGDMC contribute to enhancing its performance?
- **RQ3.** How do the hyperparameters affect the error detection performance of our KGDMC?
- **RQ4.** How does the robustness of KGDMC and other baseline methods in different noise types?

### 6.1 Experimental Setup

*6.1.1 Datasets.* In experiments, we select two widely-used real-world knowledge graph datasets: FB15K-237 [38] and WN18RR [11]. They are derived from Freebase and WordNet, respectively. Since both of these datasets are unimodal and lack visual information, we first follow [43] to augment images for all entities. Then we generate modality noise and triplet noise for each dataset in a ratio of 1:1:1 for each noise type according to Section 4. Following previous works [13, 29, 52], the ratio of total errors is set to 5%. The overall statistics of our constructed datasets are shown in Table 1.

*6.1.2 Baselines.* We compare our KGDMC framework with various competitive baselines. For modality error detection, we finetune vision-and-language models that are widely used in multiple image-text matching scenarios as our baselines, including CLIP [32], ViLT [24], ALBEF [28], and METER [15]. For triplet error detection, we

utilize the following competitive methods for comparison. The first type is the knowledge graph error detection methods, following [29], including TransE [5], DistMult [47], ComplEx [39], KGTtm [23], CAGED [51], KG-BERT [49], StAR [42], CSProm-KG [9], and CCA [29]. Another type is the multimodal knowledge graph embedding methods, including IKRL [46], RSME [43], and VISTA [27].

*6.1.3 Evaluation Metrics.* We employ ranking measures to assess the efficacy of baselines and our proposed KGDMC. Specifically, within a multimodal KG, we rank all entities and triplets based on their modality and triplet confidence scores in descending order, respectively. Entities or triplets that exhibit higher confidence scores are more prone to being erroneous. Following [29, 51, 52], we adopt precision@$K$ and recall@$K$ as evaluation metrics. Precision@$K$ represents the proportion of false entities or triplets identified among the top $K$ entities or triplets with the highest scores. Recall@$K$ denotes the percentage of correctly identified false entities or triplets relative to the total number of erroneous entities or triplets.

*6.1.4 Implementation Details.* In our experiments, we initialize the weights of BERT and ViT by leveraging the pre-trained versions: BERT-base-cased and ViT-base-patch16-224-in21k, respectively. The maximum length of the word sequence for textual input is set to 128. Furthermore, all images are resized to a resolution of $224 \times 224$, with a patch size $P$ of 16. We utilize the Adam [25] optimizer with a batch size of 128 for optimizing. The margin parameter $\gamma$ is empirically set to 1.0. Experiments were conducted on a PC with 256 GB RAM, 4 Intel(R) Xeon(R) Gold 6226R CPUs and an NVIDIA GeForce RTX A6000 GPU with 48 GB memory.

### 6.2 Experimental Results

*6.2.1 Overall Performance (RQ1).* Table 2 and Table 3 present the experimental results of our KGDMC in comparison with baselines. For modality error detection, we find that these vision-and-language baselines exhibit competitive results, especially CLIP. This benefits from their pre-training in image-text matching using large-scale image-text corpus. However, they indiscriminately consider the entities with modality errors and lack fine-grained semantic consistency mining, which limits their performance. Our KGDMC significantly outperforms these baselines and achieves the best metrics on two datasets. Specifically, when $K$ equals to 5%, it achieves 11.2% and 9.9% absolute improvement on FB15K-237 and WN18RR, respectively. This demonstrates the effectiveness of DMR module design in KGDMC for addressing modality error detection.

For triplet error detection, we have the following observations. First, traditional knowledge graph embedding models, such as TransE, DistMult, and ComplEx, perform unsatisfactorily due to their underlying assumption that all triplets are correct, thereby failing to learn discriminative representations for normal and noisy triplets. Second, text-based methods, including KG-BERT, StAR, CSProm-KG, and CCA, exhibit notably superior performance compared to structure-based knowledge graph error detection models like KGTtm and CAGED. This is attributed to the benefits of incorporating additional factual knowledge obtained from large-scale open-domain corpora through pre-trained language models, along with the detailed information in textual descriptions. Third, multimodal knowledge graph embedding models IKRL and VISTA exhibit

**Table 2: Experimental results of modality error detection. The results are averaged 5 runs using different random seeds.**

| Metrics | Precision@$K$ | | | | | | | | | | Recall@$K$ | | | | | | | | | |
|---|---|---|---|---|---|---|---|---|---|---|---|---|---|---|---|---|---|---|---|---|
| Datasets | FB15K-237 | | | | | WN18RR | | | | | FB15K-237 | | | | | WN18RR | | | | |
| $K$ | 1% | 2% | 3% | 4% | 5% | 1% | 2% | 3% | 4% | 5% | 1% | 2% | 3% | 4% | 5% | 1% | 2% | 3% | 4% | 5% |
| CLIP | 47.6 | 43.1 | 39.0 | 37.9 | 37.6 | 67.7 | 54.2 | 48.5 | 42.8 | 37.9 | 9.5 | 17.2 | 23.4 | 30.3 | 37.6 | 13.5 | 21.6 | 29.1 | 34.2 | 37.9 |
| ViLT | 29.7 | 27.2 | 26.8 | 27.0 | 26.4 | 48.2 | 38.6 | 34.2 | 31.4 | 30.6 | 5.9 | 10.9 | 16.1 | 21.6 | 26.4 | 9.6 | 15.4 | 20.5 | 25.1 | 30.6 |
| ALBEF | 42.1 | 37.6 | 36.9 | 35.1 | 34.8 | 63.1 | 50.5 | 45.7 | 40.2 | 36.1 | 8.4 | 15.0 | 22.1 | 28.1 | 34.8 | 12.6 | 20.2 | 27.4 | 32.1 | 36.1 |
| METER | 40.0 | 36.2 | 35.1 | 31.7 | 30.4 | 56.5 | 43.9 | 39.7 | 35.2 | 33.4 | 8.0 | 14.4 | 21.0 | 25.3 | 30.4 | 11.3 | 17.5 | 23.8 | 28.2 | 33.4 |
| KGDMC | **62.0** | **58.3** | **55.7** | **53.2** | **48.8** | **80.2** | **66.5** | **57.9** | **51.7** | **47.8** | **12.2** | **23.2** | **33.4** | **42.5** | **48.8** | **16.0** | **26.6** | **34.8** | **41.3** | **47.8** |

**Table 3: Experimental results of triplet error detection. We report the results of some baselines according to CCA [29].**

| Metrics | Precision@$K$ | | | | | | | | | | Recall@$K$ | | | | | | | | | |
|---|---|---|---|---|---|---|---|---|---|---|---|---|---|---|---|---|---|---|---|---|
| Datasets | FB15K-237 | | | | | WN18RR | | | | | FB15K-237 | | | | | WN18RR | | | | |
| $K$ | 1% | 2% | 3% | 4% | 5% | 1% | 2% | 3% | 4% | 5% | 1% | 2% | 3% | 4% | 5% | 1% | 2% | 3% | 4% | 5% |
| TransE | 94.6 | 77.4 | 60.6 | 49.8 | 42.3 | 69.0 | 57.6 | 50.1 | 43.7 | 40.0 | 18.9 | 31.0 | 36.4 | 39.9 | 42.3 | 13.8 | 23.1 | 30.0 | 35.0 | 40.0 |
| DistMult | 76.4 | 63.0 | 53.0 | 46.3 | 41.0 | 68.7 | 63.3 | 52.6 | 43.8 | 37.4 | 15.3 | 25.2 | 31.8 | 37.1 | 41.0 | 13.7 | 25.3 | 31.6 | 35.0 | 37.4 |
| ComplEx | 80.2 | 63.3 | 52.1 | 44.6 | 39.3 | 77.4 | 69.9 | 55.0 | 44.9 | 38.4 | 16.1 | 25.4 | 31.3 | 35.7 | 39.3 | 15.5 | 27.9 | 33.0 | 35.9 | 38.4 |
| KGTtm | 85.7 | 68.7 | 63.1 | 46.7 | 43.7 | 78.9 | 64.4 | 54.1 | 47.3 | 41.7 | 17.1 | 27.5 | 37.8 | 37.4 | 43.7 | 15.8 | 25.7 | 32.4 | 37.8 | 41.7 |
| CAGED | 86.3 | 66.6 | 60.2 | 54.3 | 46.7 | 75.3 | 62.0 | 53.6 | 47.0 | 42.1 | 17.3 | 26.6 | 36.2 | 43.5 | 46.7 | 15.0 | 24.8 | 32.1 | 37.6 | 42.1 |
| KG-BERT | 96.6 | 79.9 | 66.0 | 58.4 | 49.8 | 97.3 | 96.8 | 93.8 | 82.9 | 71.0 | 19.3 | 31.9 | 39.6 | 46.7 | 49.8 | 19.5 | 38.7 | 56.3 | 66.3 | 71.0 |
| StAR | 97.0 | **83.5** | 68.1 | 57.1 | 49.0 | 97.1 | 91.8 | 84.2 | 73.9 | 64.7 | 19.4 | **33.4** | 40.9 | 45.7 | 49.0 | 19.4 | 36.7 | 50.5 | 59.1 | 64.7 |
| CSProm-KG | 96.1 | 79.8 | 68.9 | 57.4 | 50.9 | 97.7 | 92.7 | 86.9 | 77.3 | 68.0 | 19.2 | 31.9 | 41.3 | 45.9 | 50.9 | 19.5 | 37.1 | 52.1 | 61.8 | 68.0 |
| CCA | 96.9 | 81.2 | 70.7 | 59.9 | 53.4 | 98.6 | 95.9 | 92.0 | 83.4 | 73.3 | 19.4 | 32.5 | 42.4 | 47.9 | 53.4 | 19.7 | 38.4 | 55.2 | 66.7 | 73.3 |
| IKRL | 94.7 | 76.3 | 65.2 | 56.8 | 47.4 | 95.9 | 86.9 | 77.3 | 67.7 | 60.0 | 18.9 | 30.5 | 39.1 | 45.4 | 47.4 | 19.2 | 34.8 | 46.4 | 54.2 | 60.0 |
| RSME | 96.4 | 80.3 | 67.6 | 58.2 | 49.8 | 97.2 | 92.9 | 89.8 | 78.4 | 69.3 | 19.3 | 32.1 | 40.6 | 46.6 | 49.8 | 19.4 | 37.1 | 53.9 | 62.7 | 69.3 |
| VISTA | 95.3 | 77.2 | 66.1 | 57.0 | 48.1 | 96.7 | 88.4 | 78.8 | 69.5 | 61.4 | 19.1 | 30.9 | 39.7 | 45.6 | 48.1 | 19.3 | 35.3 | 47.3 | 55.6 | 61.4 |
| KGDMC | **97.6** | 83.2 | **72.1** | **61.4** | **55.1** | **99.2** | **97.8** | **94.6** | **84.5** | **74.5** | **19.5** | 33.3 | **43.3** | **49.1** | **55.1** | **19.8** | **39.1** | **56.7** | **67.6** | **74.5** |

**Table 4: Ablation study of precision/recall at top-$K$ results for modality error detection.**

| Models | FB15K-237 | | | WN18RR | | |
|---|---|---|---|---|---|---|
| | $K$=1% | $K$=3% | $K$=5% | $K$=1% | $K$=3% | $K$=5% |
| KGDMC | **62.0/12.2** | **55.7/33.4** | **48.8/48.8** | **80.2/16.0** | **57.9/34.8** | **47.8/47.8** |
| w/o $\mathcal{L}_{DA}$ | 24.1/4.8 | 23.2/13.9 | 21.7/21.7 | 31.1/6.2 | 24.4/14.7 | 19.3/19.3 |
| w/o $\mathcal{L}_{Intra_{DD}}$ | 57.2/11.4 | 54.6/32.7 | 48.1/48.1 | 75.3/15.1 | 57.2/34.3 | 47.4/47.4 |
| w/o $\mathcal{L}_{Inter_{DD}}$ | 57.9/11.6 | 54.9/32.9 | 48.3/48.3 | 77.1/15.4 | 57.3/34.4 | 47.5/47.5 |
| w/o $z_{t_I}$ | 54.5/10.9 | 52.5/31.5 | 47.1/47.1 | 73.8/14.8 | 55.9/33.6 | 46.7/46.7 |
| w/o $C_M$ | 56.6/11.3 | 53.7/32.2 | 47.7/47.7 | 71.4/14.3 | 55.7/33.4 | 47.1/47.1 |

**Table 5: Triplet error detection results of ablation study.**

| Models | FB15K-237 | | | WN18RR | | |
|---|---|---|---|---|---|---|
| | $K$=1% | $K$=3% | $K$=5% | $K$=1% | $K$=3% | $K$=5% |
| KGDMC | **97.6/19.5** | **72.1/43.3** | **55.1/55.1** | **99.2/19.8** | **94.6/56.7** | **74.5/74.5** |
| w/o $v_e$ | 97.0/19.4 | 70.9/42.5 | 54.2/54.2 | 99.1/19.8 | 92.8/55.7 | 73.9/73.9 |
| w/o $w$ | 96.7/19.3 | 68.6/41.2 | 51.6/51.6 | 98.3/19.6 | 91.3/54.8 | 71.8/71.8 |
| w/o $S_{LT}$ | 97.2/19.4 | 71.7/43.0 | 54.7/54.7 | 98.7/19.7 | 94.1/56.4 | 74.1/74.1 |
| w/o $S_{GS}$ | 97.3/19.5 | 71.3/42.8 | 53.9/53.9 | 99.1/19.8 | 93.7/56.2 | 73.3/73.3 |
| w/o $S_{GP}$ | 97.2/19.4 | 71.1/42.7 | 53.7/53.7 | 98.6/19.7 | 93.4/56.1 | 73.4/73.4 |
| w/o $C_T$ | 96.7/19.3 | 70.1/42.1 | 53.3/53.3 | 98.1/19.6 | 91.8/55.1 | 72.2/72.2 |

inferior performance compared to text-based methods. This indicates that ignoring modality errors and directly utilizing visual information will introduce additional noise, ultimately impairing the effectiveness of triplet error detection. RSME employs a forget gate to disregard irrelevant image information, which mitigates the negative effects of modality errors to a certain degree. However, it still assumes that both entities and triplets are correct, which leads to its inferior performance compared to the state-of-the-art text-based knowledge graph error detection model CCA. Finally, our KGDMC outperforms all baselines on two datasets and achieves new state-of-the-art performance. Specifically, when $K$ equals to 5%, KGDMC gains 1.7% and 1.2% absolute improvement on FB15K-237 and WN18RR, respectively. This clearly underscores the effectiveness and superiority of our KGDMC in detecting triplet errors.

*6.2.2 Ablation Study (RQ2).* To better understand our proposed KGDMC, we conduct a series of ablation studies, as presented in Table 4 and Table 5. For modality error detection, we observe that KGDMC significantly outperforms KGDMC w/o $\mathcal{L}_{DA}$. The distribution alignment constraint serves as a fundamental cornerstone of our DMR module, constructing the modality-invariant subspaces.

Furthermore, KGDMC w/o $\mathcal{L}_{Intra_{DD}}$ and KGDMC w/o $\mathcal{L}_{Inter_{DD}}$ both result in a decrease in performance, confirming the effectiveness of these two distribution disentanglement constraints to facilitate the construction of modality-specific subspaces. We then replace $z_{t_I}$ with $z_{v_I}$ in cross-modality reconstruction, which leads to a significant performance drop. The accurate textual modality-invariant representations can better guide the reconstruction process. Finally, KGDMC w/o $C_M$ performs worse than KGDMC, emphasizing the necessity of adaptive confidence pseudo-labels to help models mitigate the negative effects of noise interference.

For triplet error detection, visual modality provides abundant complementary information, enabling KGDMC with visual modality to outperform its unimodal version KGDMC w/o $v_e$. However, the performance of KGDMC w/o $w$ significantly drops and it even underperforms the unimodal KGDMC w/o $v_e$. This is consistent with our claim that directly integrating erroneous visual information not only fails to aid in detecting triplet errors but also introduces additional noise, ultimately diminishing the overall performance. Moreover, we find that the performance experiences varying degrees of decline when separately removing each granularity of triplet confidence, confirming the effectiveness of these

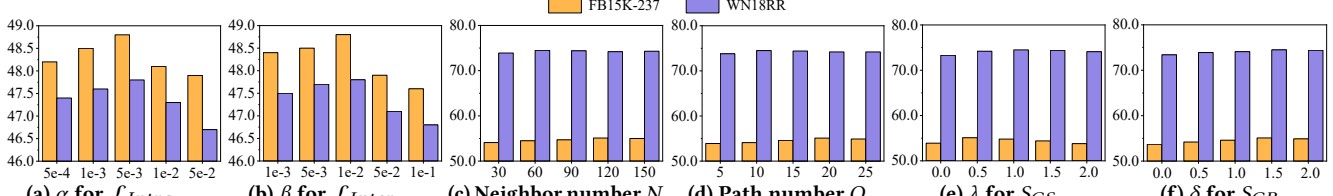

**Figure 3: The precision@5% results of our KGDMC with varying hyperparameters on FB15K-237 and WN18RR.**

**Table 6: The precision@5% results of triplet error detection on various error types.**

| | Models | TransE | DistMult | ComplEx | KGTtm | CAGED | KG-BERT | StAR | CSProm-KG | CCA | IKRL | RSME | VISTA | KGDMC |
|---|---|---|---|---|---|---|---|---|---|---|---|---|---|---|
| | Random | 72.6 | 66.2 | 73.4 | 73.0 | 75.8 | 67.4 | 72.8 | 73.2 | 76.8 | 70.3 | 72.5 | 71.1 | **77.4** |
| FB15K-237 | Similar | 30.4 | 32.8 | 30.6 | 31.8 | 33.1 | 33.6 | 37.1 | 41.9 | 45.3 | 35.5 | 41.6 | 38.7 | **50.2** |
| | Adversarial | 12.5 | 12.5 | 15.0 | 12.8 | 12.6 | 19.9 | 16.4 | 21.1 | 24.0 | 17.2 | 20.6 | 18.3 | **31.4** |
| | Random | 43.4 | 31.6 | 38.4 | 44.8 | 48.6 | 80.6 | 79.4 | 79.1 | 80.7 | 77.2 | 79.1 | 77.9 | **81.1** |
| WN18RR | Similar | 35.1 | 33.8 | 30.3 | 39.1 | 38.8 | 63.3 | 61.0 | 63.4 | 65.7 | 59.8 | 62.4 | 61.1 | **69.5** |
| | Adversarial | 31.4 | 28.5 | 36.6 | 35.9 | 37.3 | 59.9 | 52.7 | 53.6 | 54.5 | 52.8 | 54.7 | 53.3 | **62.8** |

**Table 7: Modality error detection on various error types.**

| | Models | CLIP | ViLT | ALBEF | METER | KGDMC |
|---|---|---|---|---|---|---|
| | Random | 67.0 | 49.6 | 62.4 | 59.1 | **71.9** |
| FB15K-237 | Intra similar | 34.7 | 26.0 | 30.2 | 23.6 | **50.8** |
| | Inter similar | 11.2 | 7.0 | 12.0 | 8.7 | **23.6** |
| | Random | 64.4 | 52.6 | 60.9 | 60.7 | **70.2** |
| WN18RR | Intra similar | 39.6 | 32.7 | 37.2 | 30.6 | **53.5** |
| | Inter similar | 9.8 | 6.5 | 10.3 | 8.9 | **19.4** |

three key components. They provide a variety of consistency information from different views to collaboratively detect triplet errors. Finally, similar to $C_M$ in modality error detection, the utilization of $C_T$ guides models to focus more on convincing triplets, thereby alleviating the negative effects of noisy triplets and ultimately improving overall performance.

*6.2.3 Hyperparameter Analysis (RQ3).* To investigate the influence of hyperparameters on the performance of our KGDMC, we vary them on the two datasets, as illustrated in Figure 3. For modality error detection, we can observe that KGDMC has a similar sensitivity to $\alpha$ and $\beta$ on the two datasets. A small value may render our model insufficient for effectively disentangling information, whereas an excessively large one could degrade the quality of latent variables, ultimately resulting in performance deterioration. When respectively setting $\alpha$ and $\beta$ as 0.005 and 0.01, our KGDMC gains the best results. For triplet error detection, we can find that the required number of neighbors and paths changes with different datasets, which is reasonable since WN18RR is much sparser than FB15K-237. Furthermore, increasing these numbers does not always lead to improved performance. Excessive information aggregation and passing can potentially exacerbate the negative effect of erroneous triplets. Optimal performance is achieved by our KGDMC when configured with $N = 120$ and $O = 20$ on FB15K-237, and $N = 60$ and $O = 10$ on WN18RR. Finally, $\lambda$ and $\delta$ are utilized to balance the contribution of three confidence scores for triplet error detection. Similarly, given that FB15K-237 exhibits a higher density, it tends to aggregate a greater number of erroneous triplet neighbors, which renders its $S_{GS}$ less reliable compared to WN18RR. Consequently, $S_{GS}$ has a smaller contribution on FB15K-237. When we set $\lambda$ to 0.5 for FB15K-237 and to 1.0 for WN18RR, our KGDMC obtains optimal

results. However, we find that $S_{GP}$ plays similar roles in the two datasets, and our KGDMC gains the best results when $\delta = 1.5$. This is due to the fact that when obtaining the unified path embeddings, we assign weights based on the local triplet confidence, thereby mitigating the negative impact of erroneous triplets.

*6.2.4 Perform on Different Error Types (RQ4).* To investigate the robustness of our KGDMC against various types of noise, we add 5% of each type noise to FB15K-237 and WN18RR separately. Table 6 and Table 7 show the precision@5% results. For modality error detection, while CLIP exhibits satisfactory results when confronted with random noise, it is far inferior to our KGDMC in addressing both intra-modality and inter-modality similar noise. Our KGDMC captures fine-grained semantic consistency across modalities by leveraging modality disentanglement and reconstruction techniques, enabling it to effectively identify highly similar erroneous entities. For triplet error detection, since baselines have distinguished random noise well, our KGDMC only slightly improves their performance. However, for semantic similar and adversarial noise, our KGDMC significantly outperforms all baselines. This is because the visual modality, coupled with multi-grained consistency from various views, provides additional complementary information for distinguishing these hard-to-detect noises.

# 7 Conclusion

In this paper, we are the first to introduce the novel multimodal knowledge graph error detection task, which aims to simultaneously detect both modality errors and triplet errors. To address this task, we propose a novel KGDMC framework. Specifically, we design a disentanglement modality reconstruction module to capture fine-grained semantic consistency across modalities to identify modality errors. Furthermore, we propose a multi-grained triplet confidence module to integrate the internal self-consistency of triplets, the consistency of aggregated neighbor information, and the consistency of information propagation, for triplet error detection. Experimental results on our constructed two multimodal knowledge graph datasets demonstrate the superiority of our KGDMC. In the future, we will explore the use of KGDMC to assist downstream tasks based on multimodal knowledge graphs.

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
