# OpenReview forum: "Multimodal Knowledge Graph Error Detection with Disentanglement VAE and Multi-Grained Triplet Confidence"
_ACM.org/TheWebConf/2025/Conference — WWW 2025 Poster_

### Official Review · Reviewer_yx1Z · 2024-11-11

**Novelty:** 6
**Technical Quality:** 4

**Review:**

This paper addresses a new and relevant task in multimodal knowledge graphs: error detection across modalities. The authors present a novel dataset for this task and propose an innovative method for error detection in multimodal KG.

Pros:
- The paper introduces a novel task focused on multimodal KG error detection, which is an underexplored area with significant potential for practical impact.
- The authors provide a new dataset tailored to this task, contributing valuable resources to the research community.

Cons:
- Some of the baseline models used in the evaluation are not entirely convincing. Please see specific questions below for further clarification.
- Technical Accuracy:
    - About TransE (lines 545-547): The paper claims that the plausibility scores from TransE correspond to the likelihood of a triple's correctness, which is misleading. The plausibility scores do not have probabilistic interpretation [1,2].
    - Interpretation of Distances (line 548): The distance $∣∣ℎ+𝑟−𝑡∣∣_{2}$ is defined as confidence score in the paper. This seems contrary to the fact that a smaller distance represents higher plausibility.
- Due to randomness of negative sampling of TransE or other similar KGE methods, the ranking and plausibility scores of bottom ranked triples are quite random. The approach of using bottom-ranked triples for error detection appears to introduce a degree of randomness, which raises concerns about the reliability of this method.

**Questions:**

- Could the authors clarify why confidence scores are sorted in descending order, assuming that entities and triples with higher confidence scores are more error-prone? In the referenced paper "Knowledge Graph Error Detection with Contrastive Confidence Adaption," confidence scores are ordered in ascending order, where triples with lower scores are presumed to be noisier.

- The paper mentions that TransE is employed to encode structure information. Could the authors clarify why to choose TransE but not other similar KGE methods?

- It’s unclear if knowledge graphs meet the assumptions of social community theory, especially given the diversity of relation types in KGs. Additionally, when using Graph Attention Networks (GATs) to learn node embeddings, how is relation heterogeneity handled?

- Since the plausibility scores of bottom-ranked entities might be influenced by the randomness of negative sampling in TransE, could the authors elaborate on how they address the potential impact of this randomness on the experiment results? If the bottom-ranked entities are indeed random, this could call into question the robustness of the reported results.

**Reviewer Confidence:**

3: The reviewer is confident but not certain that the evaluation is correct

**Scope:**

4: The work is relevant to the Web and to the track, and is of broad interest to the community

---

### Official Review · Reviewer_w4EQ · 2024-11-29

**Novelty:** 4
**Technical Quality:** 4

**Review:**

This paper looks at the problem of detecting errors in multimodal knowledge graphs. It proposes a new model to perform checks on the consistency of the modality with the knowledge graph and whether the triple statement is erroneous or not.

Strengths
- The problem itself is interesting and well described. The separation between modality consistency and triple errors is useful.
- The approaches to add noise are useful in particular the more specific approaches in terms of adding noise that is semantically similarity and adversarial way.
- The large numbers of comparison to existing error detection approaches for KGs and baseline models (e.g. CLIP)
- The distentagled VAE seems novel. I don't think I've seen this done before.

Weaknesses
- Only two well-worn datasets (WN18RR and FB15k-237) are used. There are a wide variety of other more up to date datasets in particular Wikidata5M or http://kgbench.info that include multimodal data.
- Missing related work in terms of current approaches to checking the quality of KGs specifically the use of constraint languages like ShEx. It's important to link to what's actually occurring in practice like in the Wikidata Schemas project (https://www.wikidata.org/wiki/Wikidata:Schemas)
- In some cases the precision of the definitions of what constitutes an error could be improved.

**Questions:**

- Why did you not use an existing multimodal KG?
- What do you mean by if an image fails to associate with an entity? What is the precise definition here.
- When you say that an entity is irrelevant? Is this with respect to a relation in terms of the domain and range definitions of the relation or is relevance defined another way.

**Reviewer Confidence:**

4: The reviewer is certain that the evaluation is correct and very familiar with the relevant literature

**Scope:**

3: The work is somewhat relevant to the Web and to the track, and is of narrow interest to a sub-community

---

### Official Review · Reviewer_HvNE · 2024-11-29

**Novelty:** 5
**Technical Quality:** 6

**Review:**

The paper deals with the problem of detecting errors in knolwedge graphs, specifically multimodal knowledge graphs that mix text and image data. Authors claim that the introduction of images pose new challenges for solving this problem, and demand revisiting older challenges.

The approach taken is to construct a model that first embeds text and images, and then uses a Disentanglement VAE to extract information that is not modality-specific from these embeddings. Then, to compute scores of triples authors suggest certain triple-specific measures, plus a graph transformer over the neighbourhood of nodes.

As for benchmarks, authors take two well known datasets, augment them with images, and then introduce random noise that model errors in the database. Their methodology is shown to be more than competitive in this dataset.

I think this paper is necessary, multimodal kg as important and so is the problem of detecting errors in KGs. But one important aspect where this needs improvement is in the writing, specifically in the conceptual part. For me, the main problem is that I don't understand the problem. Triple error can be understood as a triple classification problem, but since it must be unsupervised, the authors take on a lot of assumptions (see section 5.3) about how triples in KGs should behave, and then optimize around this. Without a proper problem definition, I cannot check whether these assumptions are part of the problem, or they are heuristics. But then, what is the goal of Section 5.2? What is a modality confidence score?

I could not understand the ablation study because I cannot follow the notation used for the column "Models". Same for Table 6 (Models) and Table 7 (Models). There are several other problems with the grammar, so you should also consider proofreading the paper a few more times.

**Questions:**

Is it true that both the Disentanglement VAE and the triple confidence module are completely independent modules?

Can you specifically define the problem that your model addresses? I'm confused between all the scores you compute.

Amongst the challenges you mention, the lack of labeled data and triplet errors are prone to happen in any KG. Why do we need specific architectures to tackle these two problems for multimodal KGs?

I'm curious about the distribution of the random noise on your benchmarks. Are you considering that noise in nodes with low degree may have different consequences than nodes with high degree? (In particular, if you divide train and test set by partitioning triples, it is likely that nodes with high degree will be always present in your training set).

**Reviewer Confidence:**

2: The reviewer is willing to defend the evaluation, but it is likely that the reviewer did not understand parts of the paper

**Scope:**

3: The work is somewhat relevant to the Web and to the track, and is of narrow interest to a sub-community

---

### Official Review · Reviewer_JEu2 · 2024-12-02

**Novelty:** 5
**Technical Quality:** 6

**Review:**

This paper focuses on multi-modal knowledge graph error detection (MKGED) which is introduced in this paper. The authors first construct two MKGED datasets, and then propose a framework KGDMC. Experimental results confirm its effectiveness.

Pros:
1. The introduced multi-modal knowledge graph error detection task is very important.
2. The constructed datasets are valuable.
3. Extensive experiments confirm the motivation.

Cons:
1. Some details, such as data quality are missing.
2. No code and dataset available.

**Questions:**

1. How to ensure the quality of the constructed FB15K-237 and WN18RR.

2. As in Figure2, why concatenate $z _ {v _ s}$ and $z _ {t _ I}$ for decoder.

**Reviewer Confidence:**

3: The reviewer is confident but not certain that the evaluation is correct

**Scope:**

4: The work is relevant to the Web and to the track, and is of broad interest to the community

---

### Official Review · Reviewer_eKkb · 2024-12-02

**Novelty:** 4
**Technical Quality:** 3

**Review:**

### **Evaluation Summary**

The paper proposes a new task — Multimodal Knowledge Graph Error Detection (MKGED), and introduces Disentanglement VAE and Multi-Grained Triplet Confidence (MTC) modules to detect both modality errors and triplet errors.


---

### **Pros:**

1. The paper is well-written and easy to follow.

2. The use of Disentanglement VAE to capture semantic consistency across modalities through cross-modality reconstruction is logically sound and has demonstrated improvements in performance.

3. The MTC improves the robustness of error detection by incorporating multiple levels of consistency.

---

### **Cons:**

1.  The use of Depth-First Search in Global path confidence to mine paths can lead to high computational complexity, particularly in large-scale knowledge graphs.

2.   Line 626 of the paper mentions that these datasets lack images. If certain entities do not have corresponding images, how did the authors perform image augmentation?

3.  In Figure 1, there appears to be a typographical error with the name "Jebron James," and in line 167, "the the cross-modality" contains a redundancy.

**Questions:**

1. What is the difference between knowledge graph error detection and triplet classification?

2. Mining paths in knowledge graphs is inherently time-consuming. How do the computational burdens associated with implementing the proposed method affect the model's training and inference speed?

**Reviewer Confidence:**

2: The reviewer is willing to defend the evaluation, but it is likely that the reviewer did not understand parts of the paper

**Scope:**

4: The work is relevant to the Web and to the track, and is of broad interest to the community